# Improved Cordycepin Production by Cordyceps Militaris Using Corn Steep Liquor Hydrolysate as an Alternative Protein Nitrogen Source

**DOI:** 10.3390/foods13050813

**Published:** 2024-03-06

**Authors:** Ying Chang, Xiaolan Liu, Yan Jiao, Xiqun Zheng

**Affiliations:** 1College of Food Science, Heilongjiang Bayi Agricultural University, Daqing 163319, China; 2College of Food and Bioengineering, Qiqihar University, Qiqihar 161006, China; 02541@qqhru.edu.cn

**Keywords:** corn steep liquor hydrolysate, cordycepin production, liquid cultivation, low cost alternative nitrogen source

## Abstract

Cordycepin production in the submerged culture of *Cordyceps militaris* was demonstrated using hydrolyzed corn processing protein by-products, known as corn steep liquor hydrolysate (CSLH), as an alternative nitrogen source. The growth, metabolism, and cordycepin production of *Cordyceps militaris* were evaluated under various concentrations of CSLH induction. The results demonstrated that CSLH addition had positive effects on the growth and cordycepin production with various *C. militaris* strains. The optimum strain, *C. militaris* GDMCC5.270, was found to effectively utilize CSLH to promote mycelium growth and cordycepin production. Low concentrations of CSLH (1.5 g/L) in the fermentation broth resulted in 343.03 ± 15.94 mg/L cordycepin production, which was 4.83 times higher than that of the group without CSLH. This also enhanced the metabolism of sugar, amino acids, and nucleotides, leading to improved cordycepin biosynthesis. The increase in key amino acids, such as glutamic acid, alanine, and aspartic acid, in the corn steep liquor hydrolysate significantly enhanced cordycepin yield. The corn steep liquor hydrolysate was confirmed to be a cost-effective accelerator for mycelium growth and cordycepin accumulation in *C. militaris*, replacing partial peptone as a cheap nitrogen source. It serves as a suitable alternative for efficient cordycepin production at a low cost.

## 1. Introduction

Cordycepin, or 3-deoxyadenosine, is a nucleoside analog and the principal active compound found in cordyceps. It possesses various hygienic and pharmacological properties, including antioxidant, antimicrobial, and antiviral activities, as well as immunomodulatory effects [1,2,3,4]. Moreover, cordycepin has been proven to inhibit adipose accumulation [5] and exhibit anti-inflammatory and antiproliferative effects on human cancer cells [6,7]. These characteristics make it a potential candidate for clinical applications in cancer treatment, such as leukemia. Furthermore, recent research has demonstrated cordycepin’s effectiveness against Coronavirus Disease 2019 (COVID-19) [8].

In addition to its efficacy as a therapeutic medicine, cordycepin possesses anti-photoaging and anti-pigmentation properties, making it a promising bioactive ingredient for cosmeceutical products. Consequently, there is a significant demand for cordycepin in the food, medicine, and cosmetics industries [9,10,11]. Regrettably, the widespread utilization of cordycepin has been hindered by its limited production efficiency and high cost.

Currently, cordycepin is primarily obtained from the metabolites of *Cordyceps militaris*. The artificial cultivation of cordyceps involves two main approaches: liquid culture and solid-state fermentation [12,13]. The submerged culture method is increasingly favored for cordycepin production due to its numerous advantages, such as higher mycelial yield in a confined space, reduced contamination risks, and efficient biological processes [14,15]. Nitrogen and carbon sources play a pivotal role in the growth of *C. militaris* and the biosynthesis of cordycepin during the liquid culture process. In particular, the availability of high-quality, cost-effective nitrogen sources is crucial for efficient cordycepin production [16].

Corn steep liquor (CSL) is a valuable by-product of the corn wet milling industry that encompasses a wide range of essential nutrients, including reducing sugars, vitamins, minerals, corn protein, and amino acids such as glutamate, glutamine, glycine, and alanine. These amino acids serve as an excellent source of organic nitrogen. CSL’s nutrient composition makes it an economical option for providing essential microbial nutrients for various applications, particularly as a complex nitrogen source for metabolite production [17]. Because of the low cost and wide availability, CSL has been used in diverse industrial fermentation processes. However, the limited utilization rate of CSL protein restricts its effectiveness as an efficient nitrogen source [18]. In our previous study, we found that CSL and hydrolyzed CSL as the only nitrogen source showed low cordycepin production, but hydrolyzed CSL could serve as an alternative nitrogen source for *C. militaris* growth, that CSL hydrolysate contains a high proportion of peptides and amino acids, and that it may be a superior nitrogen source for cordycepin production.

Currently, the challenges associated with cordycepin production stem from low efficiency and high costs, including a lengthy production cycle, low cordycepin accumulation, and weak strain viability in *C. militaris*. To enable the broad application of cordycepin in the food, pharmaceutical, and cosmetic industries, it is imperative to identify low-cost and highly effective nutrient factors for cordyceps cultivation. An effective strategy in this pursuit involves metabolic regulation of cordycepin biosynthesis and the enrichment of cultured *C. militaris* under cost-effective conditions.

Efforts have been made to develop efficient protein hydrolysates that enhance the production of active metabolic ingredients via microbial fermentation [19]. During cordycepin biosynthesis, glucose is first converted into glucose-6-phosphate (G-6-P), which then transforms into ribose-5-phosphate (R-5-P) via the pentose phosphate pathway (pp pathway). R-5-P serves as the starting material for the purine nucleotide pathway, involving a series of conversions from phosphoribosyl pyrophosphate (PRPP) to IMP and further to AMP, GMP, adenine, and adenosine, ultimately resulting in cordycepin [20]. Peptides and certain amino acids have been found to promote cordycepin synthesis via the known metabolic pathway of cordycepin. In the liquid-submerged fermentation of *C. militaris*, the addition of the antioxidant glutathione (GSH) helps regulate the cellular redox state and enhances cordycepin biosynthesis [21]. Glutamate and glycine, acting as precursors of the nitrogenous base residue in the de novo purine nucleotide pathway, are involved in nucleoside biosynthesis. Furthermore, L-alanine was found to increase cordycepin yield by activating pathways related to energy generation and amino acid interconversion [22]. Aspartate proved to be an important precursor for cordycepin production in *C. militaris*, as its addition enhanced cordycepin production [23].

CSL primarily consists of corn protein (comprising 40% to 60% of dry matter) with a rich amino acid profile, including glutamic acid, alanine, aspartate, and others. This suggests that CSL hydrolysate (CSLH) could serve as a promising culture medium for *C. militaris* due to its high amino acid content. Furthermore, small molecule peptide nitrogen sources may have a regulatory effect on cordycepin production and enrichment. Thus, CSLH has the potential to replace expensive yeast extract and peptone as a novel, cost-effective nitrogen source for *C. militaris*. Despite the available literature, there are no reports on the utilization of CSLH as a nitrogen source for the fermentation process of *C. militaris*. Therefore, the main objective of this study is to develop CSLH as a cost-effective nitrogen source for the submerged culture of *C. militaris* and to investigate the impact of CSL and CSLH on cordycepin production and mycelium biomass.

## 2. Materials and Methods

### 2.1. Materials

Corn steep liquor, with a moisture content of 61.6%, was obtained from Qiqihar Longjiang Fufeng Biotechnology Co., Ltd. (Qiqihar, China). An alkaline protease (200,000 u/g) was purchased from Novo Nordisk (Bagsvaerd, Denmark). Cordycepin (≥98%) was obtained from Sigma-Aldrich Co. (St. Louis, MO, USA). Chromatographically pure methanol was sourced from Zhenxing Chemical Reagent Co., Ltd. (Shanghai, China). Double distilled water (Milli-Q water) was utilized for the preparation of the mobile phase and all solutions. All other chemicals and reagents used were of analytical grade.

### 2.2. Preparation of Corn Steep Liquor Hydrolysate (CSLH)

The enzymatic hydrolysis of CSL was conducted using alkaline protease, following a previously published protocol with some modifications [24]. Briefly, the pH of the CSL solution was adjusted to 10.0 with a 1.0 mol/L NaOH solution. Alcalase was then added to the CSL solution at an enzyme-to-substrate ratio of 1:100 (g/g dry-weight protein). The mixture was hydrolyzed at 50 °C for 2.0 h. The hydrolysis reaction was terminated by deactivating the enzyme at 95 °C for 5 min. Subsequently, the enzymolysis liquid was freeze-dried and stored at 4 °C for later use in the culture medium. The composition of the main nutrient content between CSL and CSLH is shown in Table 1.

### 2.3. Strain and Culture Conditions

#### 2.3.1. Microorganisms

The strain of *C. militaris* GDMCC5.270 was purchased from the Guangdong Microbial Culture Collection Center (Guangzhou, China), *C. militaris* CICC14014 was purchased from the China Center of Industrial Culture Collection (Beijing, China), *C. militaris* IMASC9-3 was purchased from the Institute of Microbiology, Heilongjiang Academy of Sciences, *C. militaris* ACCC5224 originated from the Agriculture Culture Collection of China (Beijing, China), and the QFCM-1 strain was preserved in our laboratory. The stock culture was maintained on a potato dextrose agar (PDA) medium consisting of 4.0 g/L potato starch, 20 g/L glucose, and 15 g/L agar. After a 7-day incubation period at 25 °C, the inoculated slants were stored at 4 °C for subculture.

#### 2.3.2. Preparation of Spore Suspensions

The spores on the PDA slant surface were first flooded with sterile water, and the spore suspensions were harvested by scraping slants using a sterilized platinum loop. The spores were suspended in a sterile and dry 250 mL Erlenmeyer flask containing glass beads and cultured in a constant temperature oscillating incubator at 180 rpm and 25 °C for 3 h to fully disperse and activate the spores. The concentration of the spores was measured using a hemacytometer. The spore suspension was diluted to a concentration of 1.0 × 10^5^ cfu/mL by a basic culture medium.

#### 2.3.3. Submerged Fermentation of *C. militaris*

The prepared spore suspensions were added to 100 mL basic fermentation liquid medium with the inoculum size of 1.0% *v*/*v*, and the *C. militaris* was cultivated in a shaker at an agitation rate of 120 rpm, and after an 8-day incubation period at 25 °C, the fermentation broth was collected for further analysis [25].

The basic fermentation medium without CSL or CSLH as a control group is as follows: Glucose served as the carbon source, while peptone, yeast extract, beef paste, (NH_4_)_2_SO_4_, or mixed CSLH were utilized as nitrogen sources for cordyceps culture. The basic fermentation medium consisted of 10 g/L glucose, 5.0 g/L peptone (or yeast extract, beef paste, (NH_4_)_2_SO_4_), 0.6 g/L KH_2_PO_4_, and 0.5 g/L MgSO_4_, with a pH of 6.5. The CSL or CSLH group was used as an alternative nitrogen source at concentrations ranging from 0.75 to 4.5 g/L.

### 2.4. Analytical Methods

#### 2.4.1. Determination of Mycelium Dry Weight

Mycelium was washed with distilled water and dried at 100 °C overnight to determine mycelium dry weight (DCW) [26].

#### 2.4.2. Determination of Cordycepin and Adenosine by HPLC

The concentrations of cordycepin and adenosine in the fermentation broth were analyzed using high-performance liquid chromatography (HPLC) with a 260 nm UV detector (Primaide 1430, Hitachi, Japan). For the determination, a YMC-C18 column (4.6 × 250 mm, 5 μm) was employed at a constant temperature of 25 °C. The mobile phases consisted of water and methanol in a linear gradient from 5:95 (*v*/*v*) at 0 min to 20:80 (*v*/*v*) at 30 min, with a flow rate of 1.0 mL/min [27]. The cordycepin yield per gram of cell weight (Y_P/x_) and the specific rate of cordycepin production (Q_P_) were calculated using the following equations:Y_P/X_ = P/X(1)
Q_P_ = dP/Xdt (2)

Y_P/X_, the concentration of cordycepin produced by gram mycelium weight, mg/g; P, cordycepin concentration, mg/L; X, Cell dry weight, g; Q_P_, specific rate of cordycepin production, mg/g∙h.

#### 2.4.3. Amino Acid Analysis

The amino acid composition and content analysis were conducted using an amino acid analyzer (Hitachi L-8900, Hitachi, Japan) [22].

#### 2.4.4. The Glucose and Protein Assay

The reducing sugar concentrations were all analyzed by the 3,5-dinitrosalicylic acid method [28]. Total sugar concentration in the fermentation broth was assayed by the phenol-sulfuric acid method [29,30]. Total protein was determined by Kjeldahl nitrogen assay. The concentration of soluble protein was determined by the Lowry method using bovine serum albumin (BSA) as standard [31].

### 2.5. Statistical Analysis

All data obtained in this work were the means of triplicate experiments and were expressed as mean ± standard deviation. The data were analyzed using SPSS software (IBM SPSS Statistics V21.0) with one-way ANOVA and Tukey’s multiple comparison test to determine the difference among samples, and *p* < 0.05 was considered significant.

## 3. Results

### 3.1. Microbial Strain Selection for Cordycepin Production

The five cordyceps strains were subjected to cultivation using the basic fermentation medium (Control), as well as mixed CSL or CSLH fermentation medium, with an additional 2.0 g/L CSL or CSLH as an alternative nitrogen source to replace the same quality of peptone. Figure 1A,B represent the cordycepin concentration and DCW, respectively, after an 8-day culture for the five *C. militaris* strains using CSL or CSLH as an additional nitrogen source. As depicted in Figure 1A, *C. militaris* strains GDMCC5.270 and QFCM-1 displayed higher cordycepin productivity compared to the other three strains when CSL or CSLH was added to the medium. Notably, all five strains of *C. militaris* exhibited increased cordycepin productivity compared to the controls when CSL or CSLH was present, suggesting that *C. militaris* could absorb and utilize CSL or CSLH to enhance cordycepin production, with CSLH resulting in particularly favorable productivity. Furthermore, CSL and CSLH significantly promoted the growth of all five *C. militaris* strains (Figure 1B).

Table 2 illustrates the variations in growth and cordycepin production capabilities among different strains when supplemented with CSL or CSLH. Notably, strain GDMCC5.270 exhibited the highest values for Y_P/X_ (27.25 mg/g) and Q_P_ (0.142 mg/g·h) compared to the other strains. These results indicated that GDMCC5.270 possessed a superior capacity for cordycepin production when the nitrogen source was CSLH. Additionally, GDMCC5.270 demonstrated the highest rate of cordycepin production, suggesting its ability to efficiently produce cordycepin at a faster pace using a corn pulp complex nitrogen source. Consequently, GDMCC5.270 was chosen as the experimental strain for subsequent experiments.

### 3.2. Effect of Different Nitrogen Sources on Cordycepin Production

Here, we focused on the effect of different nitrogen sources on mycelium growth and cordycepin production of *C. militaris* GDMCC5.270. Figure 2A presents the cordycepin production under various nitrogen source conditions, including sole and mixed CSLH. Interestingly, beef paste and peptone, when compared to commonly used sole nitrogen sources in a microbial culture like (NH_4_)_2_SO_4_, exhibited greater effectiveness in promoting cordycepin production. Specifically, the combination of peptone and CSLH served as the most advantageous nitrogen source for cordycepin production, leading to a maximum production of 277.29 ± 18.60 mg/L. These findings indicated that the type of nitrogen source played a significant role in cordycepin synthesis.

The analysis of DCW (Figure 2B) exhibited similar outcomes, demonstrating that organic nitrogen sources, such as peptone, yeast extract, and beef paste, were more favorable for the growth of *C. militaris* GDMCC5.270 compared to inorganic nitrogen sources like (NH_4_)_2_SO_4_ when used as the sole nitrogen source. Furthermore, the introduction of CSLH into the culture medium showed enhanced mycelium growth and increased DCW compared to using sole nitrogen sources in the basic fermentation medium. This observation highlighted the crucial role of CSLH in promoting *C. militaris* growth and augmenting the cordycepin content.

Of utmost significance, the inclusion of CSLH as a partial replacement for these nitrogen sources resulted in remarkable advancements in the growth of *C. militaris* and the production of cordycepin. Notably, peptone combined with CSLH emerged as the optimal nitrogen source for further enhancing the cordycepin production process. Under this specific nitrogen source condition, *C. militaris* strain GDMCC5.270 exhibited the highest cordycepin production capacity (Y_P/x_) and production rate (Q_p_), as evidenced in Table 3. These findings indicated that the vitality of *C. militaris* was most potent when provided with this complex nitrogen source, thereby facilitating optimal cordycepin production. As a result, CSLH combined with peptone was ultimately selected as the most effective nitrogen source for cordycepin production in *C. militaris* strain GDMCC5.270.

### 3.3. Effect of CSLH Concentrations on Mycelium Growth and Cordycepin Production

We probed into the impact of CSLH concentrations on mycelium growth and cordycepin production in *C. militaris* GDMCC5.270. As depicted in Figure 3A, by supplementing the medium with CSLH at concentrations ranging from 0.75 to 4.5 g/L to replace a significant proportion of peptone (15% to 90%), an increase in cordycepin production was observed proportional to the CSLH concentration. Notably, at a CSLH concentration of 1.5 g/L, the cordycepin yield and DCW reached 343.03 ± 15.94 mg/L and 9.36 ± 0.21 mg/L, respectively, surpassing those of the control group (producing 70.97 ± 5.70 mg/L cordycepin and having a DCW of 0.71 ± 0.04 mg/L). However, cordycepin production began to decline when the CSLH concentration exceeded 1.5 g/L, and both cordycepin production and DCW started to decrease when the CSLH concentration exceeded 3.75 g/L. Therefore, our findings demonstrated that high CSLH concentrations inhibited the growth of *C. militaris* and the production of cordycepin.

Hence, it was evident that CSLH was unable to fully substitute peptone in the growth medium for *C. militaris*. Conversely, the inclusion of CSLH (1.5 g/L) satisfactorily fulfilled the nitrogen source requirements of *C. militaris*. Ultimately, the combination of CSLH and peptone emerged as a novel and effective nitrogen source for enhancing cordycepin production in *C. militaris* strain GDMCC5.270.

### 3.4. Effect of CSLH Addition on Substrate Metabolism

#### 3.4.1. Effect of CSLH Addition on Sugar Utilization

The effect of different concentrations of CSLH on the sugar utilization of the *C. militaris* GDMCC5.270 strain was investigated. Figure 3B depicts the sugar consumption at different CSLH concentrations after an 8-day culture. It is widely recognized that glucose serves as the optimal carbon source for cordycepin production [32,33]. In the basic culture medium, 10 g/L glucose was utilized, with approximately 89.04% consumption observed after 8 days of cultivation. In contrast, upon the addition of differing CSLH concentrations, a significant enhancement in the sugar consumption rate was observed. CSLH concentrations ranging from 0.75 g/L to 4.5 g/L augmented glucose utilization from 95.52% to 99.98%. Notably, 0.75 g/L CSLH yielded the maximum sugar utilization of 99.8% within the same cultivation period. These findings implied that CSLH enhanced sugar utilization and consequently increased cordycepin yield in *C. militaris*. This effect could be attributed to the provision of requisite amino acids by CSLH, thereby promoting cordycepin synthesis and overall sugar metabolism. Consequently, the growth of *C. militaris* and cordycepin production were positively influenced.

#### 3.4.2. Effect of CSLH on Amino Acid Utilization of *C. militaris*

Numerous studies have suggested that certain proteins and their enzymatic hydrolysates have the potential to enhance cordycepin yield. However, it should be noted that each nitrogen source possesses distinctive peptide and amino acid compositions. Consequently, the utilization of nitrogen sources by *C. militaris* varies, thereby yielding differing effects on cordycepin production [34].

To investigate the correlation between nitrogen source elements and cordycepin production, the amino acid constituents of CSLH and its availability in *C. militaris* were analyzed, as shown in Table 4. Table 4a illustrates that both CSL and CSLH contain a diverse range of amino acids but exhibit distinct amino acid compositions. The total amino acid ratio of CSLH increased as a result of CSL hydrolysis, with a significant augmentation in the content of most amino acids compared to CSL. Particularly noteworthy was the pronounced increase in aspartic acid (23.94%), glutamate (24.24%), glycine (30.01%), and alanine (23.47%) levels within CSLH. Aspartic acid, glutamate, and glycine were vital amino acids within the cordycepin synthesis pathway. Additionally, alanine played a crucial role in energy molecule production and contributed to the enhancement of cordycepin production via amino acid conversion [35]. Consequently, CSLH exhibited a higher amino acid content that can effectively regulate the synthesis of cordycepin. This indicated that the hydrolysis of CSL generated amino acid components that promoted cordycepin production. Thus, CSLH, serving as a nutrient-rich nitrogen source, yielded higher cordycepin production compared to CSL while simultaneously facilitating the growth of *C. militaris*.

Table 4b presents the amino acid availability of nitrogen sources in the fermentation broth of *C. militaris*, revealing a substantial alteration in amino acid proportions before and after the fermentation process. In the control group, despite peptone being an excellent nitrogen source, many amino acids exhibited reduced consumption by Cordyceps during fermentation, resulting in low amino acid availability. This suggested that amino acids derived solely from peptone nitrogen sources were not fully utilized by *C. militaris*. Conversely, in the CSLH group, fermentation led to a significant decrease in the content of 15 out of 17 amino acids. The utilization of nearly all amino acids notably increased, particularly glutamate, alanine, valine, isoleucine, leucine, and arginine. The utilization of aspartic acid was also increased on the basis of aspartate synthesis and transformation. Moreover, the total amino acid content in the fermentation broth decreased by more than 30%, indicating that CSL after enzymatic hydrolysis contains a greater proportion of amino acids that are readily absorbed and utilized by *C. militaris*, thereby promoting mycelium growth and cordycepin synthesis. This enhanced availability of amino acids in CSLH was a crucial factor contributing to the notable increase in cordycepin content.

#### 3.4.3. Effect of CSLH on Some Metabolite in *C. militaris* Fermentation Broth

Table 5 presents the analysis results of various fermentation metabolites of *C. militaris*. The addition of CSL and CSLH led to a significant increase in cordycepin, adenosine, polysaccharide, and soluble protein content in the fermentation broth, suggesting that CSL and CSLH effectively promoted the growth and metabolism of *C. militaris*, subsequently elevating the levels of cordycepin and other metabolites. Notably, CSLH exerted a notably superior effect compared to CSL. This difference could be attributed to the capacity of CSL and CSLH to stimulate the growth of *C. militaris* and generate a greater quantity of bioactive metabolites, such as polysaccharides and soluble proteins. Moreover, CSL and CSLH possessed higher levels of glycine, aspartic acid, and alanine, which are key amino acids involved in the synthesis and metabolism of hypoxanthine nucleotide and adenosine. Their presence indirectly promoted cordycepin synthesis by regulating the production of precursor substances required for cordycepin synthesis. Consequently, CSLH emerged as an effective enhancer for cordycepin synthesis.

## 4. Discussion

### 4.1. Improvement in Cordycepin Production for Different C. militaris Using CSL and CSLH

Microbial strains play a crucial role in the production of cordycepin, exhibiting varying degrees of nutrient absorption and utilization. Initially, we demonstrated the beneficial effects of CSL and CSLH on the growth of *C. militaris* and cordycepin production. Subsequently, we screened various strains and identified one that exhibited the highest utilization rate of CSL and CSLH for further fermentation experiments. The results indicated that CSLH served as an effective nitrogen source in inducing cordycepin production in *C. militaris*. Nevertheless, the utilization rates of CSLH differed among the *C. militaris* strains, with strain GDMCC5.270 displaying the highest productivity of cordycepin. This finding establishes strain GDMCC5.270 as the most suitable choice for cordycepin production when using a novel medium supplemented with CSLH as a nitrogen source. Consequently, *C. militaris* strain GDMCC5.270 was selected for cordycepin production in this study, aiming to maximize efficiency.

### 4.2. Advantages of CSLH

Numerous studies have emphasized the significance of nitrogen sources in the growth of *C. militaris* and its production of cordycepin. Typically, *C. militaris* relies on costly nitrogen sources like silkworm chrysalis meal, yeast extract, and peptone to facilitate optimal growth and achieve high cordycepin yields [36]. Nevertheless, the expense associated with conventional nitrogen sources poses a barrier to the affordability and widespread adoption of cordycepin. Consequently, the development of a cost-effective medium becomes imperative to enhance cordycepin production.

CSL has been recognized as a cost-effective alternative to yeast extract and peptone in microbial media, offering a low-cost nitrogen source. For example, corn steep liquor has been used as a low-cost nutrient source for urease production for Sporosrcina pasteurii as well as for recombinant Lactoccoccus lactis for antifreeze proteins [37,38,39]. In comparison, CSLH contains a higher concentration of small molecule proteins and functional amino acids such as glutamate, glycine, and alanine. Consequently, CSLH facilitates superior absorption and metabolism in microorganisms, making it a promising substitute for expensive nitrogen sources in industrial settings. Surprisingly, the application of CSLH as a nitrogen source in industrial fermentation processes remains limited, with no published reports available in the literature regarding its utilization for cordycepin production.

In this study, we assessed the impact of both traditional organic and inorganic nitrogen sources on the growth of Cordyceps and cordycepin production. Our findings demonstrated the superiority of organic nitrogen sources and specifically highlighted the beneficial effects of CSLH in enhancing cordycepin synthesis. Furthermore, the addition of CSLH improved the utilization of sugars and amino acids by *C. militaris*. Notably, the inclusion of CSLH led to significant increases in important metabolites within the *C. militaris* fermentation broth, such as cordycepin, adenosine, polysaccharides, and soluble proteins. Additionally, we discovered, for the first time, that the combination of peptone and CSLH effectively promoted the growth of Cordyceps and maximized cordycepin output. Interestingly, we observed that low concentrations of CSLH (0.75–4.5 g/L) were particularly conducive to cordycepin production.

### 4.3. Comparison to Other Studies

Several studies have documented optimal conditions for submerged culture to maximize cordycepin production, with results summarized in Table 6. Glucose and peptone have been consistently identified as beneficial carbon and nitrogen sources for cordycepin production in *C. militaris*. Furthermore, auxiliary nitrogen sources such as silkworm chrysalis powder, corn steep powder, and casein hydrolysate have demonstrated a significant impact on cordycepin production. In our study, we utilized CSLH as a mixed nitrogen source in the medium (10 g/L glucose, 3.5 g/L peptone, and 1.5 g/L CSLH) to enhance cordycepin production. The content of cordycepin reached an impressive 343.03 ± 15.94 mg/L, with an average productivity of 42.88 mg/L∙day, surpassing most reported culture conditions. The integration of CSLH reduced peptone consumption, increased cordycepin production, and shortened the cultivation time.

Our findings indicated that CSLH effectively promoted cordycepin generation as a viable alternative nitrogen source, outperforming corn steep powder and even silkworm chrysalis powder in terms of cordycepin yield. Moreover, CSLH facilitated higher cordycepin production within a shorter incubation time (8 days). Consequently, CSLH has the potential to serve as a high-quality nitrogen source for robust cordycepin production compared to previous studies.

### 4.4. Schematic Illustration of the Biosynthetic Pathway of Cordycepin Affected by CSLH

The biosynthesis pathway of cordycepin is outlined in Figure 4, with key amino acids such as aspartate, glutamic acid, glycine, alanine, and valine regulating its production. Building upon previous research and the cordycepin synthesis pathway, we proposed a preliminary explanation for CSLH’s ability to promote high levels of cordycepin accumulation. CSLH contained an abundance of small molecular amino acids, including glutamine, glycine, alanine, and peptides, which facilitated the absorption and metabolism of *C. militaris*. These components played a crucial role in the metabolism of sugars, amino acids, and nucleotides, ultimately increasing the biosynthesis of inosine monophosphate (IMP) and promoting the synthesis of IMP and adenosine, the precursors of cordycepin.

The biosynthesis pathway of cordycepin is described in Figure 4. Cordycepin was produced by the regulation of some key amino acids, such as aspartate, glutamic acid, glycine, alanine, and valine. Based on the above research and the synthesis pathway of cordycepin, the possible mechanism of CSLH promoting a high level of cordycepin accumulation was preliminarily illustrated. CSLH was rich in glutamine, glycine, alanine, and other small molecular amino acids and peptides, which were conducive to the absorption and metabolism of *C. militaris* and played a key role in sugar, amino acid, and nucleotide metabolism, increasing the biosynthesis of inosine monophosphate (IMP), and promoting the synthesis of IMP and adenosine, the precursor of cordycepin synthesis. These findings verified the essential role of CSLH in enhancing cordycepin production via gene regulation within these pathways.

Consequently, CSLH stimulated glucose and amino acid metabolism and facilitated nucleotide anabolism. As a result, the rate of cordycepin synthesis was accelerated, leading to increased cordycepin yield. Furthermore, the addition of CSLH significantly enhanced mycelium growth and the production of metabolites in *C. militaris*.

## 5. Conclusions

In this study, we confirmed that CSLH had the potential to serve as a substrate for cordycepin production in *C. militaris*. We successfully developed a mixture of organic nitrogen sources composed of CSLH, which effectively promoted the growth and cordycepin production in *C. militaris*. CSLH contained specific amino acids that were easily absorbed and utilized by *C. militaris*, regulating the metabolism of the fungus and the key genes involved in cordycepin synthesis. This regulation ultimately led to an increase in cordycepin accumulation. The utilization of low-cost corn steep liquor hydrolysate, derived from the by-product of corn wet milling processing, allows for high yields of cordycepin production. This offers a new and effective approach for industrial cordycepin production via liquid fermentation.

## Figures and Tables

**Figure 1 foods-13-00813-f001:**
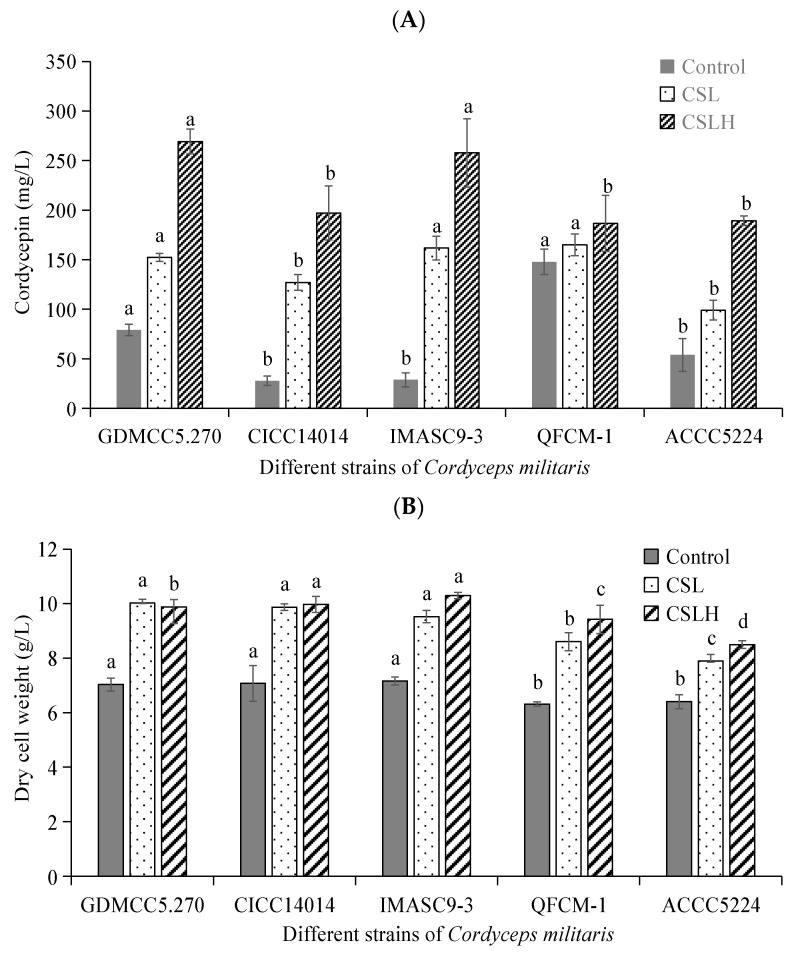
Effect of CSL and CSLH on cordycepin production (**A**) and mycelium growth (**B**) of *C. militaris*. The culture was performed at determined conditions (pH 6.5 at 25 °C and 120 r/min for 8 days). Different lowercase letters (a, b, and c) indicate that different experimental groups have significant differences (*p* < 0.05) in the same strain. Each error bar represents 1 standard deviation.

**Figure 2 foods-13-00813-f002:**
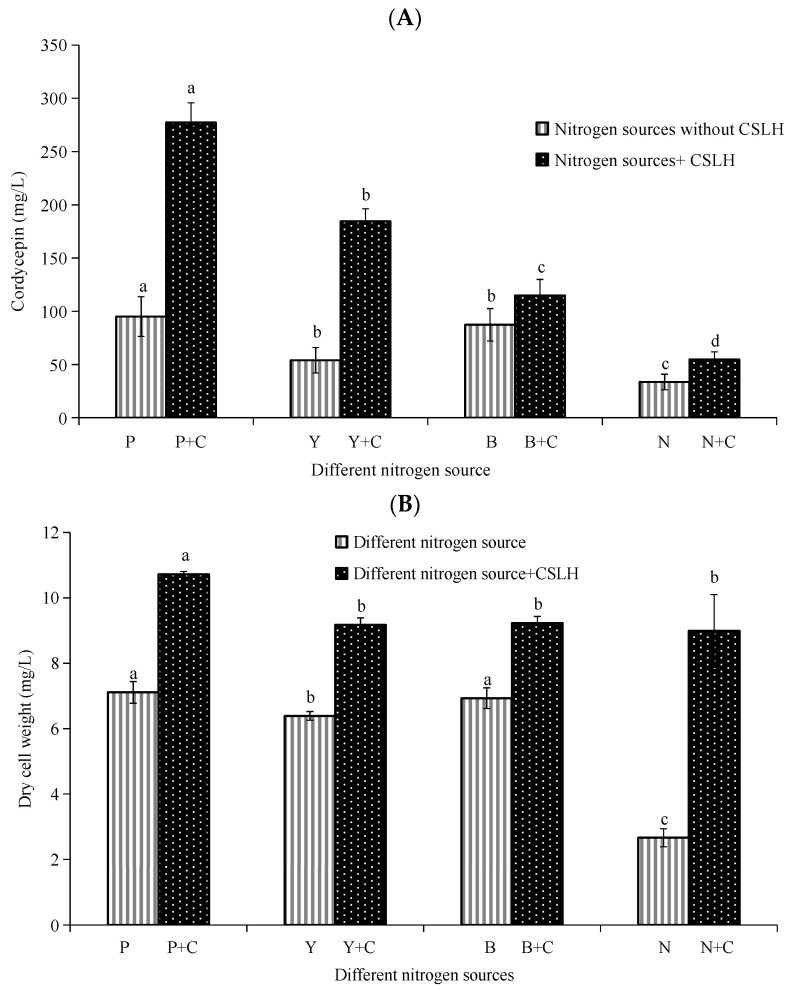
Effect of different nitrogen sources on cordycepin production (**A**) and mycelium growth (**B**) of *C. militaris* GDMCC5.270 (P: peptone 5.0 g/L, Y: yeast extract 5.0 g/L, B: beef paste 5.0 g/L, N: (NH_4_)_2_SO_4_ 5.0 g/L. P + C: peptone 3.0 g/L + CSLH 2.0 g/L, Y + C: yeast extract 3.0 g/L + CSLH 2.0 g/L, B + C: beef paste 3.0 g/L+ CSLH 2.0 g/L, N + C: (NH_4_)_2_SO_4_ 3.0 g/L + CSLH 2.0 g/L. The culture was performed at determined conditions (pH 6.5 at 25 °C and 120 r/min for 8 days). Different lowercase letters (a, b, and c) indicate that the same experimental groups have significant differences (*p* < 0.05) in different nitrogen sources. Each error bar represents 1 standard deviation.

**Figure 3 foods-13-00813-f003:**
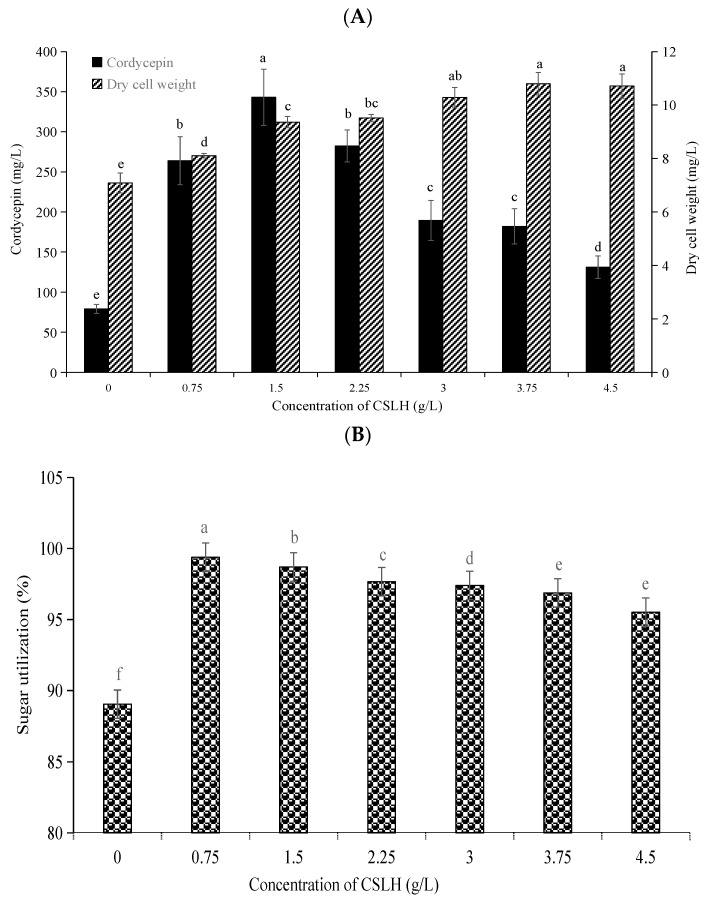
Effect of CSLH concentration on cordycepin production, mycelium growth, and sugar utilization by *C. militaris* GDMCC5.270 (**A**,**B**). The fermentation medium contained 10 g/L glucose, 5.0 g/L mixed nitrogen sources (0.75~4.5 g/L corn steep liquor hydrolyzate and 0.5~5.0 g/L peptone), 0.6 g/L KH_2_PO_4_, and 0.5 g/L MgSO_4_. Bars followed by different letters are significantly different from each other at *p* < 0.05, according to Tukey’s HSD. Each error bar represents 1 standard deviation.

**Figure 4 foods-13-00813-f004:**
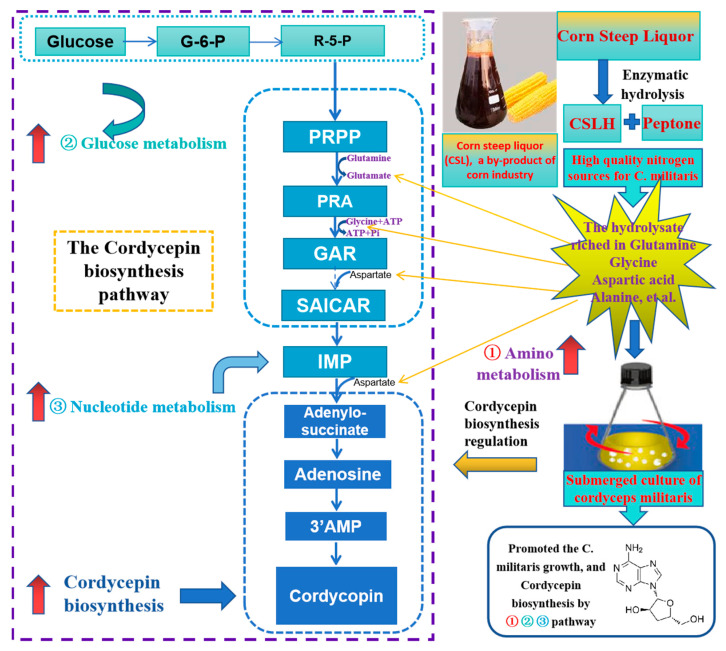
The possible mechanism of CSLH promoted the cordycepin production and mycelium growth of *C. militaris*.

**Table 1 foods-13-00813-t001:** The composition of main nutrient content between CSL and CSLH.

Main Nutrients	CSL	CSLH
Total protein (%, *w*/*v*)	21.35 ± 0.89	21.57 ± 1.25
Soluble protein (%, *w*/*v*)	10.45± 0.52	15.99 ±1.34
Total sugar (%, *w*/*v*)	2.67 ± 0.23	2.93 ± 0.18
Reducing sugar (%, *w*/*v*)	0.62 ± 0.04	0.65 ± 0.03

Values are means with standard deviations (*n* = 3).

**Table 2 foods-13-00813-t002:** Effect of CSLH on the cordycepin production, dry mycelium weight, Y_P/x_, and Q_P_ of the strains.

Strains	Cordycepin (mg/L)	Dry Cell Weight (g/L)	Y_P/x_ (mg/g)	Q_P_ (mg/g∙h)
GDMCC5.270	269.21 ± 12.52	9.88 ± 0.27	27.25	0.142
CICC14014	197.15 ± 27.26	9.97 ± 0.29	19.77	0.103
IMASC9-3	257.80 ± 34.44	10.30 ± 0.11	25.70	0.134
QFCM-1	186.6 ± 28.37	9.43 ± 0.52	19.79	0.103
ACCC5224	189.32 ± 14.77	8.51 ± 0.14	22.30	0.116

**Table 3 foods-13-00813-t003:** Effect of different nitrogen sources on the cordycepin production, dry mycelium weight, Y_P/x_, and Q_P_ of the strains.

Different Nitrogen Sources	Cordycepin (mg/L)	Dry Cell Weight (g/L)	Y_P/x_ (mg/g)	Q_P_ (mg/g∙h)
Yeast extract	54.03 ± 1.46	6.39 ± 0.14	8.46	0.044
Beef paste	87.33 ± 1.46	6.93 ± 0.31	12.60	0.066
(NH_4_)_2_SO_4_	33.56 ± 1.19	2.66 ± 0.28	12.62	0.066
Peptone	95.01 ± 4.36	7.11 ± 0.33	13.36	0.070
Yeast extract + CSLH	184.33 ± 11.93	9.17 ± 0.22	20.10	0.105
Beef paste + CSLH	114.75 ± 10.23	9.23 ± 0.20	12.43	0.065
(NH_4_)_2_SO_4_ + CSLH	54.58 ± 10.23	8.99 ± 0.11	6.07	0.032
Peptone + CSLH	277.29 ± 7.29	10.72 ± 0.17	25.87	0.135

*Y*_P/x_ = the cordycepin produced by gram mycelium weigh, mg/g; *Q*_P_ = specific rate of cordycepin production, mg/g∙h. Values are means with standard deviations (*n* = 3).

**Table 4 foods-13-00813-t004:** (a) Free amino acid types and concentrations in CSL and CSLH (%). (b) Effect of CSLH on Amino acid availability of *C. militaris* fermentation broth.

(a)						
Amino Acid		Amino Acid Concentration (g-Amino Acid/100 g-Nitrogen Source)				Increase Proportion (%)
		CSL		CSLH		
Aspartic acid		1.34		1.66		23.94
Threonine		0.68		0.87		28.30
Serine		0.72		0.94		31.34
Glutamate		3.04		3.78		24.24
Glycine		1.24		1.61		30.01
Alanine		3.58		4.42		23.47
Cysteine		0.09		0.14		68.54
Valine		1.41		1.70		20.83
Methionine		0.36		0.73		105.94
Isoleucine		0.87		1.11		27.73
Leucine		2.36		2.80		18.61
Tyrosine		0.41		0.50		20.68
Phenylalanine		1.12		1.25		11.25
Lysine		0.73		0.80		10.17
Histidine		0.69		0.61		−11.80
Arginine		0.64		0.84		31.70
Proline		2.40		2.76		15.22
Total		21.65		26.52		22.49
**(b)**						
**Amino Acid**	**Amino Acid Concentration of fermentation broth (g-Amino Acid/100 g-Substrate)**					
	**Peptone**			**Peptone + CSLH**		
	**Before Fermentation**	**After Fermentation**	**Amino acid availability** **(%)**	**Before Fermentation**	**After Fermentation**	**Amino acid availability** **(%)**
Aspartic acid	1.61	3.00	−86.72	1.65	1.96	−19.12
Threonine	0.60	0.65	−8.67	0.72	0.62	14.09
Serine	0.89	1.08	−21.09	0.854	0.73	14.22
Glutamate	2.91	3.01	−3.46	2.84	1.73	39.10
Glycine	5.54	6.38	−15.07	3.59	2.83	21.21
Alanine	2.52	2.40	4.75	2.31	1.16	49.60
Cysteine	0.012	0.013	11.7	0.03	0.048	−38.71
Valine	0.87	0.82	6.11	1.01	0.59	41.83
Methionine	0.03	0.16	−478.15	0.25	0.22	11.97
Isoleucine	0.50	0.39	22.61	0.59	0.3	49.79
Leucine	1.13	0.79	29.69	1.38	0.6	57.01
Tyrosine	0.48	0.49	−3.16	0.52	0.36	29.54
Phenylalanine	0.97	0.75	22.54	0.812	0.65	20.21
Lysine	1.09	1.11	−2.43	0.95	0.69	26.61
Histidine	0.55	0.41	26.46	0.32	0.30	5.54
Arginine	1.95	1.21	37.99	15	0.57	62.18
Proline	2.24	2.26	−0.91	2.14	1.58	26.08
Total	23.87	24.9	−4.32	21.45	14.93	30.4

**Table 5 foods-13-00813-t005:** Effect of CSLH on metabolite composition of *C. militaris* fermentation broth.

Metabolite Composition	Samples of Control Nitrogen Sources	Samples with Mixed CSL Nitrogen Sources Addition	Samples with Mixed CSLH Nitrogen Sources Addition
Cordycepin (g/L)	70.97 ± 5.70	152.20 ± 4.03	343.03 ± 15.94
Adenosine (g/L)	3.88 ± 0.01	5.92 ± 0.10	25.11 ± 1.42
Polysaccharide (g/L)	0.36 ± 0.01	0.70 ± 0.02	0.88 ± 0.01
Soluble protein (g/L)	0.77 ± 0.02	0.91 ± 0.03	1.21 ± 0.12

**Table 6 foods-13-00813-t006:** Summary of submerged culture conditions of *C. militaris* for cordycepin production.

Strain	Culture Time (Day)	Carbon Source (g/L)	Nitrogen Source (g/L)	Cordycepin Production (mg/L)	Cordycepin Productivity (mg/L·day)	Ref.
*Cordyceps militaris*	17	Glucose, 40 g/L	Yeast extract and peptone 1:1, 10 g/L	245.8	14.46	Mao and Zhong, 2006 [32]
*Cordyceps militaris*	6	Glucose, 20 g/L	casein hydrolysate, 20 g/L	445.0	74.17	Lee et al., 2019 [34]
*Cordyceps militaris*	20	Glucose 40 g/L	peptone (10 g/L)	596.59	29.83	Fan et al., 2012 [25]
*Cordyceps militaris*	18	Glucose, 42.0 g/L;	Peptone (15.8 g/L)	345.4	19.2	Mao et al., 2005 [33]
*Cordyceps militaris*	11	Sucrose, 30 g/L;	silkworm chrysalis powder, 20 g/L; yeast extract, 10 g/L;	42.53	3.87	Luo et al., 2020 [40]
*Cordyceps militaris*	12	Glucose, 40 g/L	Corn steep powder (CSP), 10 g/L;	135.0	11.2	Shih, Tsai, and Hsieh, 2007 [38]
*Cordyceps militaris*	8	Glucose, 10 g/L	Peptone 3.5 g/L, CSLH 1.5 g/L.	343.03	42.88	This study

## Data Availability

The original contributions presented in the study are included in the article, further inquiries can be directed to the corresponding author.

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
