# Peer review of "Improved Cordycepin Production by Cordyceps Militaris Using Corn Steep Liquor Hydrolysate as an Alternative Protein Nitrogen Source"

_foods, 2024, doi:10.3390/foods13050813_

Round 1
Reviewer 1 Report
Comments and Suggestions for Authors
The paper is well written and very interesting. It should provide a good base for industrial application.
My criticism lies with the abstract. L 15 “original” implies that the strain had been changed. The abstract does not make this clear that five strains were evaluated and compared to a "control".
The authors’ main objective was to evaluate CSLH in the culture of C. militaris was accomplished as stated. It would be valuable to add that in so doing various strains were evaluated as well along with various analyses. This inclusion needs to be added to the abstract and this addition in the text would prepare the reader for what follows.
It would be valuable to add the reason behind the choice of corn steep liquor. Also, why was the conversion to the CSLH made or necessary?
The word “seed (s)” line 118ff is bothersome. I don’t know if the authors mean spores, mycelium or the cultures used to inoculate the cultures for the actual experiments. Clarity is required.
In lines 111 ff cultures were purchased. Were these mycelial or spore cultures.
The word “cell” is widely used throughout the manuscript in reference to Cordyceps and data collected. Are the cells – mycelium, spore or ?? Please clarify.
In the culture of C . militaris ll124 ff, what was the volume of the liquid? The inoculum concentration? Repeatability is at stake.
Statistical methods do not include the number of replications of each fermentation strain. Were they all incubated at the same time or do we have batches of experiments. Both repeatability and statistical analysis are at stake depending on experimental design. These are not present in section 2.5. I read “all measurement … in triplicate” as the analysis of each component was performed three times on each sample of the liquid etc. or is the replications of the experiment. Not clear. why SPSS stats?
In reading through the manuscript, the authors give the impression that multiple experiments were run on the five strains and varying nutrient base. For example, wording such as “then”, “our next investigation” are used. This requires clarification.
LL 166-167 should be in the introduction regarding the purpose.
The control description is in the result section. A statement as to what it was should be in the material section.
Figure 1 etc – first the font is very small if it is to be printed. Hopefully the final will be larger or clearer. (I can enlarge on computer). For me figures should stand alone, ie. Details on statistics (does the bar represent SD or SEM?), the number of reps.
The conclusion, as well as the introduction, discuss “low-cost”. Some examples of the cost savings would be in order.
Reviewer 2 Report
Comments and Suggestions for Authors
The paper presents an interesting topic of a possible improvement in the production of cordycepin. I find the results interesting, although I do not agree with all the authors' interpretations. In my opinion, many questions remain unanswered.
1. Can CSLH be used as the sole source of nitrogen? From the data presented, I do not think so, but it would be good to have this information. If the authors have explored this possibility, it would be good to share the results; if not, the authors should explain why not.
2. The influence of CSL and CSHL on the growth and cordycepin production of the different strains is so planned that the authors’ conclusions are not fully proven. As I understand it, an amount of 2 g/l of CSL and CSLH was added to a basic medium. However, this means that the concentration of organic nitrogen sources was increased. Are the authors sure that similar results could not be obtained with other organic nitrogen sources? Please check table 2, the IMACS9-3 values seem to be wrong. It is true that the experiment with partial substitution of carbon sources by CSL and CSLH (Chapter 3.2) shows that CSL and CSLH increase both growth and cordycepin production, but Figure 1 should also show a comparison of growth with a higher concentration of organic nitrogen sources (e.g. by adding 2 g/l peptone as a second control). The authors decided to run the experiments for 8 days. Can this decision be explained? For the experimental part and the results this is not a problem, but in the discussion the authors report results from other works where the incubation time was much longer. Optimising a fermentation process involves shortening the time, but the authors should explain why they suggest an 8-day fermentation and on what basis this might be the best cost-benefit ratio.
3. The authors highlight higher glucose consumption as one of the effects of adding CSL and CSLH. Higher glucose consumption is expected when fungal growth is increased. High glucose consumption with high cell growth is not automatically proof of higher efficiency in sugar metabolism.
4. In the chapter on the amino acid composition of the culture media before and after fermentation, the aspartic acid results should be addressed.
5. Please provide more information on the methods. How was the inoculum prepared (did you count the spores? What was the volume of the flasks and liquid cultures?)
I think this paper and its results are interesting, but some changes should be made to the structure of the paper and the discussion.
Comments on the Quality of English LanguageThe English is correct, some minor changes should be made
Round 2
Reviewer 1 Report
Comments and Suggestions for Authors
The authors provided a certificate that the text had been reviewed and edited for English. This is MUCH appreciated prior to review! The revisions do not show that the revisions were put through the same scrutiny. And so, I will make some English suggestions on the additions.
L 13 – two “showed”s. Better = “The results demonstrated that CSLH addition had positive ….”
L 13 – change ‘of’ to ‘with’
L15 and following - the use of “mycelium” clarifies the whole document! Thank you!
L 56 – delete ‘nowadays’.
L 60 – place “that” as follows: “growth, that CSL”
L 60 – delete ‘indicated’ and add ‘and’.
L 93 -96 change to read ‘ militaris and to investigate … biomass’.
L 95 – delete “aimed … concentrations.”
L 128 – change ‘infiltrated’ with ‘flooded’
L 131 – change ‘in’ to ‘at’
L 132 - change 'spore’ to ‘spores’
L 136 – delete ‘then’
L 136 - change 'into’ to ‘to’
L 140 – delete ‘, it was composed’ to ‘is’
Section 3.1
2.5 – statistical analysis. I am still uncertain as to the experimental design.
It looks like there are four sets of experiments. I would tell the reader this.
Section 3.1 results --- … does this mean that for each strain and its conditions (control, CSL and CSLH) there were 3 flasks (repetitions) for each condition? This would mean that there would be 9 flasks for strain A, 9 for strain B, etc. So … in one experiment there were 45 flasks (5 strains x 9).
How were these flasks arranged on the shaker? I presume randomized, a CRD. However, this is NOT stated.
The text says ’in duplicate’. So …was the complete experiment of 45 flasks repeated in either time or space, i.e. one month later, for example, or using two shakers at the same time? If this is true, then then the statistics must include time or space as a variable in the analysis. The text needs to reflect the analysis, and this is carried over into each figure. If the experiment was replicated, i.e. repeated over time or space, then each data point mean does not reflect 3 values but 6, assuming no significance due to replication. It follows then that the captions with each figure should indicate such.
All this is to say that I doubt if the conclusions will change. However, for clarity of scientific presentation the experimental design and its statistics must be clear otherwise some doubt overshadows the conclusions.
For section 3.2 and use of strain GDMCC5.270
Is the design for this the same as for the strains? I realize authors said ‘ALL’ but …
For section 3.3 and CSLH and GDMCC5.270
Is the design for this the same as for the strains? I realize authors said ‘ALL’ but …
Sections 3.4 (.1 and .2)
Is the design for this the same as for the strains? I realize authors said ‘ALL’ but …
As I said in my initial review that each figure/table should stand alone. For me as a reader, I should not have to go back digging through the text for details.
L 202 Figure 1 . Reword as follows: “Bars followed by different letters within the same strain are significantly different from each other at p<0.05, according to Tukey’s HSD”. Further, some statement needs to be made like “Each error bar represents 1 standard deviation” (I guess???)
L 229 Figure 2 – edit accordingly: “Bars followed by different letters within the same group are significantly different from each other at p<0.05, according to Tukey’s HSD”. Further, some statement needs to be made like “Each error bar represents 1 standard deviation” (I guess???)
Table 1 - the +/- = standard dev, SEM?? on how many samples??
Table 2, 3 - +/- = standard dev, SEM??
Figure 3. authors CANNOT use the same edited caption as earlier figures. There are NO groups. Edit accordingly “Bars followed by different letters are significantly different from each other at p<0.05, according to Tukey’s HSD”. Further, some statement needs to be made like “Each error bar represents 1 standard deviation” (I guess???)
Tables 4-1 and 4-2. Is this an experiment? All other tables represented show a SD (presumably). Why not here? NO statistics?? Why?
LL 378 – 382. Edit according to: source. “For example, corn steep liquor has been used as a low-cost nutrient source for urease production for Sporosrcina pasteurii as well as for recombinant Lactoccoccus lactis for antifreezing proteins.”
Comments on the Quality of English Languagesee comments to authors
